# On the Low Degree of Entropy Implied by the Solutions of Modern Macroeconomic Models

**DOI:** 10.3390/e24121728

**Published:** 2022-11-25

**Authors:** Ragnar Nymoen

**Affiliations:** Department of Economics, University of Oslo, Blindern, P.O. Box 1095, 0315 Oslo, Norway; ragnar.nymoen@econ.uio.no

**Keywords:** macroeconomics, entropy, causal solution, rational expectations, dynamic stability, New-Keynesian Phillips curve, DSGE models, A12, B41, C50, E12, E13, E31

## Abstract

The non-causal (“forward-looking”) solution used routinely in academic macroeconomics may represent a violation of a law of entropy, namely that the direction of time is one way (from the past and towards the present), and that the variance of economic processes increases with time. In order to re-establish a degree of compatibility with the law of entropy, so called hybrid forms are required add-ins to DSGE (Dynamic Stochastic General Equilibrium) models. However, the solution that uses hybrid forms is a particular special case of a causal solutions of autoregressive distributed lags, VARs and recursive and simultaneous equations models well known from empirical macro econometrics. Hence, hybrid forms of small scale DSGE models can be analysed and tested against competing model equations, using an econometric encompassing framework.

## 1. Introduction

Cross-disciplinary examinations of economics have started to emerge; see, for example, ref. [1] (neuroscience) and [2] (physics), as well as [3,4] from within economics. As noted by [5], who writes from the perspective of neuroscience, these analyses have implications for macroeconomic theories, in particular for how decision making is represented in macroeconomic models. One important aspect of decision making is related to expectations formation about future outcomes of variables that are important in agents’ valuation systems. If expectations are formed in humans’ brains mainly through computation of similarities (correlations can be a special case), macro models that are solved conditional on history can be more defensible than models that invoke the assumption that agents form anticipations by working out the mathematical expectations, ultimately from assumed knowledge about future probability distributions, cf. [6,7].

In time series analysis and in dynamic econometrics, the solution of stochastic difference equations from historically given initial conditions is known as the causal solution of a dynamic model, cf. ([8], Ch. 3.1) It might be noted, in passing, that the term “causal” used in this context does not correspond in any way to the definitions of causality used in the literature about causal inference in economics, see [9,10,11,12] among others.

The causal solution covers single-equation autoregressive distributed lags (ADLs), VARs, and recursive and simultaneous equation models. However, in academic macroeconomics, a completely different solution, which is called the “forward-looking solution”, has become the dominant solution type. It is closely associated with the abovementioned theory of model consistent rational expectations, which is a particular special case of expectations formation mainly by the use of logic, rather than by use of similarities and heuristics.

There is some danger of creating an artificial distinction here, as research have showed that rational expectations (in the original meaning of unbiased forecasts on average) can be achieved by learning [13,14]. However, macroeconomic DGPs are characterized by non-stationarities that are due to intermittent location shifts (breaks) which may significantly hamper the possibility of learning fast enough to avoid repeated forecast failures, cf. [15,16] ([17], Ch. 12). However, these research questions go beyond the scope of this paper.

There are other inter-disciplinary issues with forward-looking solutions as well. In this note, we comment on the possible inconsistency of this solution type with the thermodynamic laws that physicists have, for a long time, regarded as the laws that apply most basically to the real world. The second law specifies that heat will not pass spontaneously from a hotter to a colder body without some change in the system. Or, as a generalization, that the process of heat conduction cannot be completely reversed at any finite cost. Besides forbidding the construction of perpetual motion machines, the second law defines *entropy*: Because energy dissipates as heat whenever work is carried out, heat cannot be collected back into useful, organized form. The evolution of the universe will be characterized by increased randomness, also known in physics as disorder. The universe is expanding faster than it can equilibrate.

It may be of interest for students, but also for developers and users of macroeconomic models, to note that the law of increasing disorder (entropy) means that the universe is one way and not reversible. The second law is therefore the expression in physical form of what we call time. Models that imply that the economy can run equally well forward or backward may therefore violate the law of entropy.

However, care must be taken, since the second law (entropy is non-decreasing with time) only holds for closed systems. Open systems can be characterized by limited (or controlled) internal entropy. In economics, this can give rise to the question about whether the degree of entropy implied by solutions of models is relative to the status of models as open (in practice, with exogenous random variables) or closed (systems of equations). The distinction between open and closed models is well established in economics, and is important in particular with regard to identification and estimation.

In an operational environment in which a model’s solutions is used for forecasting, the distinction between open and closed becomes less clear. The reason is that the future values of the exogenous variables in an open model will be unknown and will have to be forecasted as well. In principle, and often in practice, we can think of those forecasts as the solutions of a set of equations for the exogenous variables, so the model used for forecasting is, in effect, a closed model.

As noted above, there is another conceptual distinction central in time series econometrics, which is between causal and non-causal solutions of a dynamic equation (closed as well as open). A causal solution fundamentally depends only on knowledge about initial conditions, and is thus consistent with the idea that time runs one way, from history and to the present. A non-causal solution runs the other way, from the distant future and backwards to the present. Mathematically, the two types of solutions work equally well. For example, subject only to the assumption of no “unit-roots”, both solutions generate random variables that are covariance stationary time series ([17], Ch. 3). Nevertheless, time series statisticians usually have little patience with the non-causal solution, and remarks such as “irrelevant” or “without practical interest” are not uncommon in those disciplines ([8], p. 81). Relevant exceptions exist, however. Recently, progress has been made in the use of non-causal solutions of autoregressive processes with explosive roots that generate bubbles; see [18].

One area of model building in which the non-causal solution has obtained a strong foothold is in macroeconomics. Over the last few decades, academic macro economists developed a strong taste for non-causal solutions. DSGE models in particular seem to depend on that class of solutions. DSGE models are manifestations of the microeconomic hegemony in economics, which has resulted from the “New Classical Counter Revolution (NCCR)”, a term attributed to Wren-Lewis [19,20].

Returning to the second law of thermodynamics, macrodynamic models that are characterized by causal solutions are at least weakly compatible with the law of entropy. It can be remarked that the stationary version of the causal solution most likely overstates the tendency of evolving economies to reach steady states (equilibria). The “degree of compatibility” is, however, higher for models that have causal solutions that contain stochastic trends (due to unit-roots at the zero-frequency), which has become standard in the development of modern empirical econometric macro models.

The issues that we point out follow directly from the use of non-causal solutions to derive the evolution paths of endogenous variables of DSGE models. Non-causal solutions imply that the variables become independent of the historical evolution of the system. A typical macro economic model equation characterized by so-called pure forward-looking behaviour has this property, which strongly refutes the second law of thermodynamics. Unsurprisingly, producers of DSGE models have been forced to replace the “pure” forward-looking behaviour with hybrid forms, which re-install an element historical dependency in the final solution, but which are completely ad hoc when judged against the principles laid down by the leaders of NCCR. Nevertheless, as we shall see, the future dependency is clearly also present in the solution of the hybrid form.

In this note, we show for simple models that the root of the violation has to do with the implementation of model consistent rational expectations in DSGE models, for all or some of the economic agents. It is beyond the scope of this paper to cover the different modifications that have disseminated from the NCCR, but we venture to believe that the main thrust of the argument is also relevant to models in which learning provides an asymptotic justification for the rational expectations hypothesis, cf. ([13], Ch. 1.4) [21,22] among others.

## 2. Causal and Non Causal Solutions

The future dependency of solutions that are typical of DSGE models can be shown by the use of a simple dynamic model equation for a time series $y_{t}$(t=0,±1,±1,…):(1)$$y_{t}=a_{0}+a_{b}y_{t-1}+e_{t},,$$
$a_{0}$,$a_{b}$ are parameters and $e_{t}$(t=0,±1,±1,…) is a second time series independent of $y_{t}$, as well as of $y_{t-j}$ (lags) and $y_{t+j}$ (leads).

However, before stepping into the territory of future dependency we can, as a reference, define what we mean by a causal solution. It is determined by an initial condition $y_{0}$, considered to be known, and by the whole history of the random process $e_{t}$(t=1,2,…). After repeated substitution (backwards in time) to the initial condition, we may write it as:(2)$$y_{t}=a_{0}\sum_{i=0}^{t-1}a_{1}^{i}+\sum_{i=0}^{t-1}a_{b}^{i}e_{t-i}+a_{b}^{t}y_{0}$$
If $e_{t}$ was a deterministic sequence, for example $e_{t}=e^{*}$ (a fixed number) for all *t*, (Equation 2) is the solution of a linear difference equation. The solution has the property of global asymptotic stability in the case where the autoregressive parameter $a_{b}$ is between −1 and 1, and it is unstable if $a_{b}=\pm 1$.

The interpretation where $e_{t}$ is a random process is more interesting for empirical macroeconomic modelling, as the difference equation in this case generates another random process, namely $y_{t}$(t=1,2,…). In the simplest case, we can take $e_{t}$ as a white noise process, with constant expectation zero and a fixed variance ($\sigma_{e}^{2}$), $-1<a_{b}<1$ represents the stationary case, since (Equation 2) “delivers” $y_{t}$ (t=1,2,…) as a stationary time series with unconditional (steady-state) expectation and variance that do not depend on time. Nevertheless, as the system evolves from t=1, to t=2, t=3 and so forth, its variance (entropy) is indeed increasing, and the time independent unconditional variance only characterizes the stationary state. Just as an ecosystem attains a climax, economic system can reach a steady state [23].

$a_{b}=\pm 1$ is the integrated case, or unit-root case. Following custom, $a_{b}=1$ defines a random-walk process, and the casual solution is then the sum of a deterministic trend ($a_{0}t+y_{0}$) and a stochastic trend $\sum_{i=0}^{t-1}e_{t-i}$. Consequently, the variance of $y_{t}$ is a monotonously increasing function of time in this case.

Within the framework of causal processes, $\left|a_{b}\right|>1$ gives rise to explosive dynamics. The solution will not be stationary: the importance of the initial conditions, for example, will grow exponentially with time, instead of declining. Such solutions are called explosive for obvious reasons. Explosive solutions are very relevant in economics, and maybe more so when they have some form of check imposed or built into the process, i.e., a mechanism that stops the explosive process, at least for a period.

The above definitions and sub-categories of a causal solution (i.e., stable/stationary and unstable/non-stationary) generalize to model equations with higher-order dynamics, and to systems of such equations. There can also be linear combinations of non-stationary variables, which define variables that are stationary, which is the property of cointegration. The causal solution of such a system requires that the associated characteristic roots are located inside or on the unit-circle; there are no roots outside the unit-circle (the generalization of $\left|a_{b}\right|>1$). This has proven to be an adaptable framework for the development of empirical macro econometric models More flexibility and generality can be achieved through specification of non-linear functional forms, and by opening up for location shifts (random Markov-switching or IIS), and volatility components (ARCH effects).

However, it was the phenomenon of future dependency that we wanted to take a closer look at. To introduce it, we can imagine ourselves being interested in answering the question: Can there be a stationary solution of (Equation 1) in the case of $\left|a_{b}\right|>1$, which we just referred to as the explosive case?

The answer to these questions is yes. To find the solution, we re-normalize (Equation 1), so that $y_{t}$ depends on the lead variable $y_{t+1}$ instead of the lag:(3)$$y_{t}=\frac{-a_{0}}{a_{b}}+\frac{1}{a_{b}}y_{t+1}+\frac{1}{a_{b}}e_{t}$$
and consider repeated forward insertion on the right-hand side in this process, giving:(4)$$y_{t}=\frac{-a_{0}}{a_{b}}\sum_{i=0}^{h-1}{\left(\frac{1}{a_{b}}\right)}^{i}+{\left(\frac{1}{a_{b}}\right)}^{h}y_{t+h}+\frac{1}{a_{b}}\sum_{i=0}^{h-1}{\left(\frac{1}{a_{b}}\right)}^{i}e_{t+i}$$
after h−1 substitutions. Clearly, since $0<a_{b}^{-1}<1$ (assuming positive $a_{b}$ for simplicity), this solution implies that the unconditional expectation and variance of $y_{t}$ is independent of time. Hence, the time series generated by (Equation 1), is indeed a stationary process.

The challenge represented by (Equation 4) is instead one of relevance. The future dependence of the solution seems to make it necessary to know the whole sequence of future shocks, and for a finite horizon *h*, the terminal condition $y_{T+h}$. In many disciplines, these requirement simply seems unrealistic, and the non-causal solution is therefore often dismissed as unpractical or “unnatural” ([8], p. 81). However, as mentioned, progress has been made in the use of non-causal solutions of processes which generate bubbles in financial markets [18]. That said, and keeping in mind that (Equation 1) is a closed system, it appears that (Equation 4) does not respect the second law of entropy, as the direction of time has been reversed.

## 3. The Non-Causal Solution in DSGE Models

As noted above, academic macroeconomics after the NCCR stands out as a field where the non-causal solution principle has thrived. As a result, non-causal processes play a large role, particularly in DSGE models, which have become something of an industry standard.

The reason is that from the start, DSGE models took on board a specific interpretation of rational expectations, namely that the economic agents form mathematical expectations about the future values of the model’s variables by utilizing the equations of the model. Said differently, households and firms behave like (some) economic forecasters do and use the optimal point forecast, which minimizes the mean squared forecast error (assuming that the model is a close approximation to the real-world data generation process (DGP), and that the DGP of the future is the same as the DGP of the past).

A structural equation that appears frequently in macro theories is (see e.g., [24], Ch.8):(5)$$y_{t}=a_{0}+\underset{>0}{a_{f}}E_{t}\left[y_{t+1}\right]+\underset{\geq 0}{a_{b}}y_{t-1}+bx_{t}+\varepsilon_{t},$$
where the difference from the earlier equations is that we have defined the variable $e_{t}$ as the sum of one observable time series variable $x_{t}$, and one unobservable variable $\varepsilon_{t}$:$$e_{t}=x_{t}+\varepsilon_{t}$$
and that we have introduced a new parameter, $E_{t}\left[y_{t+1}\right]$, which is the conditional expectation of the variable $y_{t+1}$, given the information available for forecasting it at the end of period *t*.

Setting $a_{b}=0$ would be the case of a pure forward-looking model equation as derived from microeconomic theory including rational expectations. It is a defining trait of the canonical monetary policy model [21].

$a_{b}>0$ (the positive sign does not represent any loss of generality) defines the *hybrid form*, and it is this version that dominates in DSGE models that are used in practice. The motivation is that, for reasons that are ad hoc, e.g., “sluggishness in the data” ([25], p. 15), “enhance the empirical coherence of the model” ([26], p. 104), “account for inertia in the data” ([21], p. 2066), the preferred model consistent expectations theory only becomes partly implemented. For example, when the variable $y_{t}$ represents inflation, the theoretically correct specification would be to set $a_{b}=0$, but the hybrid form with $a_{b}>0$ is nevertheless used in the DSGE model with the suggested interpretation that a proportion of the firms follows indexation rules, or other rules of thumb, rather than being fully rational price setters.

Similar “softening up” of the theoretical core specification is used for the model equations representing other parts of the macroeconomy. In the consumption and savings department, for example, a proportion of the households are allowed to be credit rationed, or for other reasons place excessive emphasis on today’s or yesterday’s wage income.

### 3.1. The Pure Form of the Model

Nevertheless, the pure form, with $a_{b}=0$, is useful for seeing how the non-causality of the solution is affected by the fact that instead of $y_{t+1}$ on the right-hand side of in (Equation 3), we have $E_{t}\left[y_{t+1}\right]$ in
(6)$$y_{t}=a_{0}+a_{f}E_{t}\left[y_{t+1}\right]+bx_{t}+\varepsilon_{t}.$$
In order to find a solution, we need to specify the information that the agents base their expectations on. We can follow custom and assume that $\varepsilon_{t}$ is a white-noise process, so that $\varepsilon_{t+i}$(i=1,2,…) cannot be predicted from (end of) period *t* information, while $x_{t}$ follows the autoregressive process
(7)$$x_{t}=c_{x1}x_{t-1}+\varepsilon_{xt},$$
where $-1<c_{x1}<1$, and $\varepsilon_{xt}$ is a white-noise process, independent of the structural disturbance $\varepsilon_{t}$.

Following ([27], pp. 100–110) the solution can be attained in two steps:(1)Find $E_{t}\left[y_{t+1}\right]$.(2)Solve for $y_{t}$.

Step 1. makes use of the non-causal solution (forward recursion) and agents assumed ability to form expectations based on the equations of the model.

We commence by re-writing (Equation 6) by using the expectation error, defined as:$$u_{t+1}=y_{t+1}-E_{t}\left[y_{t+1}\right],$$
to obtain:(8)$$y_{t}=a_{0}+a_{f}y_{t+1}+bx_{t}+\varepsilon_{t}-a_{f}u_{t+1}.$$
Identical equations hold for $y_{t+i}$ (i=1,2,…,h):(9)$$y_{t+i}=a_{0}+a_{f}y_{t+i+1}+bx_{t+i}+\varepsilon_{t+i}-a_{f}u_{t+i+1},$$
and iterating forward leads to the expressions for $y_{t}$ (only simplified by setting $a_{0}=0$)
(10)$$y_{t}=a_{f}^{h}y_{t+h}+\sum_{i=0}^{h-1}\left(a_{f}^{i}bx_{t+i}+a_{f}^{i}\varepsilon_{t+i}\right)-\sum_{i=1}^{h}a_{f}^{i}u_{t+i},\;$$
The same type of (non-causal) solution is found for $y_{t+1}$. When we take account of the process (Equation 7) that governs $x_{t}$ and $x_{t+1}$, it becomes
(11)$$y_{t+1}=a_{f}^{h}y_{t+1+h}+\sum_{i=0}^{h-1}\left(a_{f}^{i}bc_{x1}x_{t+i}+a_{f}^{i}\varepsilon_{xt+1+i}+a_{f}^{i}\varepsilon_{t+1+i}\right)-\sum_{i=1}^{h}a_{f}^{i}u_{t+1+i}$$
and represents all the information that the agents have available for forecasting $y_{t+1}$ optimally. Hence, by applying the conditional expectation, we get
(12)$$E_{t}\left[y_{t+1}\right]=a_{f}^{h}E_{t}\left[y_{t+1+h}\right]+bc_{x1}\sum_{i=0}^{h-1}a_{f}^{i}E_{t}\left[x_{t+i}\right],$$
since conditional on period *t*, the expectation of the three last sums in (Equation 11) are zero. Moreover, from knowledge of the process (Equation 7), we can find how the agents specify expectations about $x_{t}$, $x_{t+1}$, $x_{t+2},\ldots ,x_{t+h-1}$:$$\begin{array}{ccc} E_{t}\left[x_{t}\right] & = & x_{t}\\ E_{t}\left[x_{t+i}\right] & = & c_{x1}^{i}x_{t},\;i=1,2,\ldots , \end{array}$$
and (Equation 12) therefore becomes:(13)$$E_{t}\left[y_{t+1}\right]=a_{f}^{h}E_{t}\left[y_{t+1+h}\right]+bc_{x1}\sum_{i=0}^{h-1}{\left(a_{f}c_{x1}\right)}^{i}x_{t}.$$
We next follow custom and impose on the solution that the expectation horizon extends towards infinity, h→∞.
(14)$$E_{t}\left[y_{t+1}\right]=\lim_{h\to \infty }a_{f}^{h}E_{t}\left[y_{t+1+h}\right]+bc_{x1}\sum_{i=0}^{\infty }{\left(a_{f}c_{x1}\right)}^{i}x_{t}$$
Subject to the two assumptions:(15)$$\begin{array}{ccc} -1 & < & a_{f}<1 \end{array}$$(16)$$\begin{array}{ccc} -1 & < & a_{f}c_{x1}<1 \end{array}$$
the optimal forecast for $y_{t+1}$ therefore becomes
(17)$$E_{t}\left[y_{t+1}\right]=\frac{bc_{x1}}{1-a_{f}c_{x1}}$$
The second step in the solution algorithm is quickly carried out in this case: Substitution for $E_{t}\left[y_{t+1}\right]$ in the structural Equation (Equation 6), and keeping in mind that we have simplified by $a_{0}=0$, gives the solution for $y_{t}$ as
(18)$$y_{t}=\frac{b}{1-a_{f}c_{x1}}x_{t}+\varepsilon_{t},$$
showing that all dynamics have been purged from the solution, despite starting from the dynamic formulation in (Equation 6) (or the formulation with forecast error (Equation 8)). Hence, this way of modelling expectations, as agents who are rational forecasters and who know the true DGP, now and in the future, is “taking away” dynamics—reducing the degree of entropy of the system.

Written in solution equation form, the system is seen to consist of (Equation 7), determining $x_{t}$, and (Equation 18), which gives $y_{t}$ given $x_{t}$. This is a particular special case of the recursive nature of closed-form solutions, for small or medium-scale DSGE models ([26], p. 172). It means that even though (Equation 18) is static, hence the there is no dynamic response to a “y-shock”, there are implied dynamic responses in $y_{t}$ to a shock elsewhere in the system, namely in $\varepsilon_{t}$ in the marginal model equation for $x_{t}$ (Equation 7).

Since the process for $x_{t}$ is an integral part of the system, we can guess that dynamics in the *y*-equation can re-emerge by allowing higher-order dynamics in that process. For concreteness, we use second-order dynamics:(19)$$x_{t}=c_{x1}x_{t-1}+c_{x2}x_{t-2}+\varepsilon_{xt},$$
and by re-calculating $E_{t}\left[x_{t+i}\right]$ we obtain the solution:(20)$$y_{t}=\frac{b}{1-a_{f}\left(c_{x1}+c_{x2}\right)}x_{t}+\frac{ba_{f}c_{x2}}{1-a_{f}\left(c_{x1}+c_{x2}\right)}x_{t-1}+\varepsilon_{t}$$
which is a distributed lag (DL) model equation, known from econometric text books, cf. ([28], Ch. 7).

In DSGE models, this theory has been applied to both wage and price setting under the name Calvo-style price setting or New-Keynesian Phillips Curve (NKPC) [29,30,31].

There are special applications and interpretations that can be mentioned. In the price NKPC, where $y_{t}$ is the rate of change of the general price level, $x_{t}$ has been measured by the wage-share, the output-gap (log GDP as deviation from trend) and the unemployment rate (in which case one would set b<0). In some presentations, notably the influential article by [32], the disturbance term is omitted, which suggests a stronger interpretation which is often referred to as the NKPC holding in “exact form”. In the wage NKPC, $x_{t}$ has been attempted measured by a wage pressure indicator, i.e., combining variables that one think can proxy households’ wage bargaining power and firms’ ability to pay [33].

In both applications, it is seen that that a nominal price/wage shock which lasts for one period affects the nominal wage/price change in the same period as the shock occurs. There are no dynamic responses to such a shock. Since $x_{t}$ follows a dynamic process, there will, however, be dynamic inflation responses to an impulse to $\varepsilon_{xt}$. This is also true when the solution equation for $y_{t}$ is the static equation in (Equation 18).

Nevertheless, it is widely recognized that a DSGE model cannot really survive with so very constrained dynamics, and they are therefore filled up with the hybrid forms mentioned above.

### 3.2. The Hybrid Form of the Model

As an important special case, the hybrid form of (Equation 6) is
(21)$$y_{t}=a_{0}+a_{f}E_{t}\left[y_{t+1}\right]+a_{b}y_{t-1}+bx_{t}+\varepsilon_{t}.$$
where $a_{b}>0$, and the simplest expressions for the solution is obtained if we assume $a_{f}+a_{b}<1$. It is important to note that the extended model equation does not follow from the same microeconomic theory as is used to derive (Equation 6). From that perspective, (Equation 21) is an ad hoc softening of the theoretical core.

The solution for $y_{t}$, given (Equation 21) and (Equation 19) is derived by following the same two steps as above. It is:(22)$$y_{t}=r_{1}y_{t-1}+\frac{b}{a_{f}r_{2}}k_{x1}x_{t}+\frac{b}{a_{f}r_{2}}k_{x2}x_{t-1}+\frac{1}{a_{f}r_{2}}\varepsilon_{t}$$
where $r_{1}$ and $r_{2}$ are the two roots of
$$r^{2}-\frac{1}{a_{f}}r+\frac{a_{b}}{a_{f}}=0,$$
and we assume (with only a little loss of generality) that $r_{1}<1$ and $r_{2}>1$, see ([34,35] for details. The expressions for $k_{x1}$ and $k_{x2}$ are found in Appendix A.

Although the notation is somewhat cluttered, we can recognize (Equation 22) as an ADL model equation, well known from dynamic econometrics. Hence, despite the future dependency in the structural model equation representation (Equation 5), the solution of the hybrid version of the model produces an ADL model equation for inflation which is causal and conditional on $x_{t}$. In practice, (Equation 22) can be estimated by OLS, and the results compared with competing models for the focus variable, for example of inflation.

In fact, the point that the forward solution implies restrictions on the stochastic difference equations that time series econometricians normally will use is more general. Hence, for example, Ref. [36] proposed a new evaluation methodology of small-scale DSGE models based on sequential tests conducted in a Vector Autoregressive (VAR) model. This approach contributes to earlier important literature about the empirical assessment of contesting model of expectations formations models; see [37,38,39] among others. Since, for example, the New-Keynesian Phillips curve reviewed in the next section was a new model when it was presented, a testing phase of the models encompassing capabilities could have been expected. That did not happen, and the model’s success was instead largely by default, because it was micro-founded, and because had a good fit (but in hybrid form, which no longer has a clear micro foundation).

For large-scale DSGE models, and for models with lead-in-variables more generally, the closed form solution is not practical. Therefore, a method often referred to as the Fair–Taylor algorithm is often used; see [40]. Although several versions exist, the main method is to loop repeatedly through a solution sample of finite length, treating the past and terminal conditions as fixed. Hence, this method makes the non-causal nature of the solution even clearer than the closed form solution does. In practice, though, by using extended solution paths, the algorithms are designed to make the solution path of interest independent of the terminal conditions. Hence for example, if the solution period of interest is between $T_{1}$ and $T_{2}$, the algorithm adds *k* periods after $T_{2}$. The sub-path between $T_{1}$ and $T_{2}$ has to be independent of the terminal condition. As pointed out by [41], this can be achieved by pushing the terminal condition sufficiently far into the future.

One way of deciding a terminal condition is, with reference to the correspondence principle, to solve the deterministic and static version of the dynamic model. In the case of the hybrid model Equation (Equation 21), the terminal condition such determined becomes:$$y_{T_{2}+k}^{\mathrm{term}}=\frac{a_{0}}{1-a_{f}-a_{b}},$$
which is identical to the unconditional expectation of $y_{t}$ (long-run mean). Whether the marginal model equation for $x_{t}$ is (Equation 7) or (Equation 19) does not matter, as long as they are specified without constant terms. Still, the sketched algorithm is only directly applicable in deterministic form. Stochastic simulation of macro econometric models with future dated variables still appear to be no common task [41].

## 4. Empirical Illustration

In this section, we illustrate the estimation of the closed-form solution for the US inflation dataset in the seminal article by [32]—GG hereafter. The endogenous variable $y_{t}$ is the quarterly inflation rate (measured as the change in the natural logarithm of the GDP deflator) and $x_{t}$ is the (natural logarithm) of the wage share. GG presented several estimates of $a_{f}$ and $a_{b}$. ${\hat{a}}_{f}$ is typically larger than ${\hat{a}}_{b}$, and the sum was very close to unity, cf. Table 2 in GG.

Using a typical pair of estimates, (${\hat{a}}_{f},\;{\hat{a}}_{b}=\left(0.62,0.36\right)$, the implied numbers for the roots associated with (Equation 21) are ${\hat{r}}_{1}=0.54$ and ${\hat{r}}_{2}=1.07$.

However, when (Equation 22) is estimated on the same dataset, we obtain:$$\begin{array}{ccc} y_{t} & = & \begin{array}{c} 0.8732\\ {}^{\left(0.043\right)} \end{array}y_{t-1}\;+\;\begin{array}{c} 0.06879\\ {}^{\left(0.036\right)} \end{array}x_{t}\;-\;\begin{array}{c} 0.04608\\ {}^{\left(0.030\right)} \end{array}x_{t-1}\\ & \; & \;+\;\begin{array}{c} 0.0012\\ {}^{\left(0.0004\right)} \end{array}\\ \; & \; & \mathrm{OLS},\;1960(1)-1997(4),\;\mathrm{Number}\;\mathrm{of}\;\mathrm{observations}:\;152\\ \; & \; & \mathrm{Robust}\;\mathrm{standard}\;\mathrm{errors}\;\left(\mathrm{HACSE}\right)\;\mathrm{in}\;\mathrm{brackets}\;\mathrm{below}\;\mathrm{the}\;\mathrm{estimates} \end{array}$$
where the coefficient of the lag $y_{t-1}$ is 0.87, which is considerably larger than the 0.54 that the theory predicts.

There are several possible explanations of this puzzle. It may be a bias in the OLS estimates of $r_{1}$, even if the hybrid model holds exactly in the data (i.e., we would not have estimated the true parameter, even if we started from the true DGP). The well-known Hurwicz bias goes in the opposite direction (it underestimates the true parameter), and in any case, the size of that bias is normally much lower than the gap between 0.87 (estimated) and 0.54 (hypothesized if the hybrid model is correct) ([17], Ch. 4).

However, we can investigate the possibility that estimation bias is the explanation by conducting a Monte Carlo experiment.

The results are summarized in Figure 1, which plots the sequence of ${\hat{r}}_{1}$ estimates (with 95% confidence bound) that we would expect to obtain if the hybrid estimated by GG was a good empirical model of US inflation. The plot shows that the bias is small for the initial sample with 50 observations, and that it is reduced as the sample is extended.

Hence, the explanation of the difference between theoretically derived parameter (0.54) and estimated parameter (0.87) is likely to lie elsewhere. The existing literature has investigated several other econometric issues, but the possibility of severe mis-specification has also received empirical support; see ([34], Ch. 7) and [42], among others. Specifically, the underlying theory does not allow for unanticipated shifts, although crises, breaks and regimes shifts are relatively common in real economies. Ref. [42] demonstrated that potentially spurious outcomes can arise when location shifts are not modelled and rational expectations are, in fact, irrelevant.

## 5. Discussion

A reasonable summary of our assessment may be that the consistent implementation of the second law of thermodynamics would at some point become incompatible with the solutions of macroeconomic models that have disseminated from the NCCR. This is because a necessary step in the solution is to solve the model forward in time, and then introduce the principle of model consistent rational expectations as a way of making the non-causal solution practical, in terms of observable variables. The result is a solution in the form of either a static model equation, or a model that has a finite distributed lag in the forcing variable.

When the theoretical version of the model is put to the side, and is replaced by the hybrid model equation, the solution takes the form of the well-known ADL model equation. Viewed from that angle, the hybrid form gives a solution which has much in common with the type of empirical model equation that a time series econometrician would take as a natural starting point for an empirical modelling project.

However, to rationalize the ADL model by “softening up” the rational expectations core is like fording the river to fetch water. The ADL model equation, the VAR, recursive systems and simultaneous equations models, are all well founded in the theory of time series analysis and dynamic econometric modelling. Hence, we can just as well say that the ADL model equation, and its multiple-equation counterparts, represent models that are consistent with expectations that are based on experience by recognition of similarities. At the same time, they are robust to any pockets of model-based rational expectations behaviour that might exist.

More generally, to assess the empirical relevance of their theory equations, DSGE modellers should find it useful to apply the embedding check proposed by [43], retaining their equation and testing it against alternatives. Hendry and Johansen showed that when the theory is complete and correct, the theory model can be retained after selection with parameter estimates that have the same distributions as when the model is directly estimated without selection. Hence, there is little danger of valuable empirical models being lost for macroeconomics by testing any specific theory-driven model equation against both location breaks as well as other theoretical views, including those that drill deeper into the implications of expectations and of how they are formed; see [14,20,44,45] among others.

## Figures and Tables

**Figure 1 entropy-24-01728-f001:**
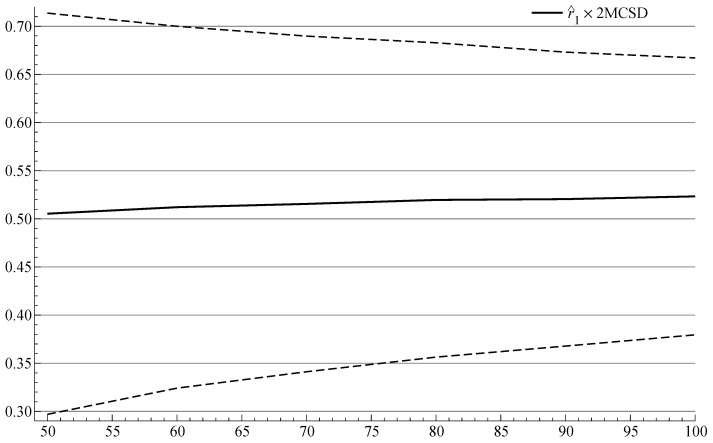
Monte Carlo simulation of recursive OLS estimation of the autoregressive coefficient in Equation (Equation 22), with ±2standarderrors (MCSD) indicated by the distance between the dashed lines, assuming that the hybrid model estimated by GG is the DGP. The true parameter is 0.54. Results based on 1K replications.

## Data Availability

Not applicable.

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
