# Peer review of "On the Low Degree of Entropy Implied by the Solutions of Modern Macroeconomic Models"

_entropy, 2022, doi:10.3390/e24121728_

Round 1

Reviewer 1 Report

This study is an excellent survey that allows physicists and other readers unfamiliar with economic models to gain a better understanding of the solution methods and philosophy of economic dynamic models. This is because, as described in this study, both of the rational expectations equilibrium solution method, which is not used in physics, and the hybrid solution method, which combines these solutions with the causal solution method, which is often used in physics, are well organized. 

Main Comments

1.    This study defines “causality” in terms of time transitions from the past to the future; on the other hand, “causal inference” occupies a central position in contemporary econometrics literature. Since the definition of causality by Imbens & Rubin (2015) or Pearl (2009), which is the basic literature in the field of “causal inference”, is likely to differ from the definition in this study, this note should be inserted in the introduction or footnotes of this manuscript. 

2.    It would be easier for readers who are not familiar with economic models to understand if section 3 were separated into a section on “the non-causal solution in DSGE models” and a new section explaining about “the hybrid form of in DSGE models “. 

3.    The leading empirical studies using macrodynamic models with hybrid forms in DSGE models are Christiano et al. (2005) and Smets & Wouters (2007). It would be more valuable as a survey of this study if these references were cited in the latter part of Section 3 or Section 4.

 References

[1] Imbens & Rubin (2015) Causal Inference for Statistics, Social, and Biomedical Sciences, Cambridge University Press.

 [2] Pearl (2009) Causality, 2nd edition, Cambridge University Press.

 [3] Lawrence J. Christiano, Martin Eichenbaum and Charles L. Evans (2005) “Nominal Rigidities and the Dynamic Effects of a Shock to Monetary Policy,” Journal of Political Economy, Vol. 113, No. 1 (February 2005), pp. 1-45

 [4] Smets, Frank, and Rafael Wouters. (2007). "Shocks and Frictions in US Business Cycles: A Bayesian DSGE Approach." American Economic Review, 97 (3): 586-606.

Author Response

See the attached pdf file.

Reviewer 2 Report

Please see attached revision report.

Author Response

See the answers in the attached pdf.
